# Virtual Breathalyzer: Towards the Detection of Intoxication Using Motion Sensors of Commercial Wearable Devices

**DOI:** 10.3390/s22093580

**Published:** 2022-05-08

**Authors:** Ben Nassi, Jacob Shams, Lior Rokach, Yuval Elovici

**Affiliations:** Department of Software and Information Systems Engineering, Ben-Gurion University of the Negev, Beer-Sheva 8410501, Israel; nassib@post.bgu.ac.il (B.N.); liorrk@post.bgu.ac.il (L.R.); elovici@bgu.ac.il (Y.E.)

**Keywords:** intoxication detection, wearable devices

## Abstract

Driving under the influence of alcohol is a widespread phenomenon in the US where it is considered a major cause of fatal accidents. In this research, we present Virtual Breathalyzer, a novel approach for detecting intoxication from the measurements obtained by the sensors of smartphones and wrist-worn devices. We formalize the problem of intoxication detection as the supervised machine learning task of binary classification (drunk or sober). In order to evaluate our approach, we conducted a field experiment and collected 60 free gait samples from 30 patrons of three bars using a Microsoft Band and Samsung Galaxy S4. We validated our results against an admissible breathalyzer used by the police. A system based on this concept successfully detected intoxication and achieved the following results: 0.97 AUC and 0.04 FPR, given a fixed TPR of 1.0. Our approach can be used to analyze the free gait of drinkers when they walk from the car to the bar and vice versa, using wearable devices which are ubiquitous and more widespread than admissible breathalyzers. This approach can be utilized to alert people, or even a connected car, and prevent people from driving under the influence of alcohol.

## 1. Introduction

In 2013, a death from a motor vehicle accident caused by an alcohol impaired driver occurred every 51 min, a tragic statistic that represents more than 30% of all US traffic-related deaths that year [1]. The high rate of fatal accidents resulting from “driving under the influence” (DUI) reflects the devastating effects of alcohol consumption on driving (e.g., reduced coordination, difficulty steering, and reduced ability to maintain lane position and brake appropriately).

Given the potential consequences of drunk driving, there would be value in an intoxication detection method that provides immediate results and is non-invasive/easy to administer without the need for expensive, specialised equipment. Currently, intoxication can be detected via a blood or breath test, such as the breath alcohol concentration (BrAC) test, which measures the weight of alcohol present within a certain volume of breath [2]. This test is conducted with a breathalyzer device [3] and uses an individual’s breath as a specimen/sample. However, these options are not practical for an individual interested in checking their own level of intoxication, as such tests can only be performed in dedicated labs or require the use of specialized equipment that may require prior calibration and ongoing maintenance. As a result, detecting intoxication based on ubiquitous devices is a scientific gap, and there is a need for a different type of test based on such devices that can be applied by an individual in real time.

In this paper, we present Virtual Breathalyzer, a new approach for the detection of intoxication based on motion sensors of smartphones and wrist-worn devices. It is a known fact that alcohol consumption causes changes in people’s movements. We hypothesize that these changes can be measured using the motion sensors of smartphones and wrist-worn devices in order to detect intoxication with a trained machine learning model. In order to assess the performance of our suggested approach, we conducted a field experiment in which we collected 60 free gait samples from patrons of three bars, using a smartphone (Samsung Galaxy S4) and wrist-worn device (Microsoft Band), and labeled the data based on the results of an admissible breathalyzer used by the police. We trained machine learning models to predict if an individual is intoxicated based on their free gait and analyzed the performance of our approach with different BrAC thresholds of intoxication and combinations of features/device sensors. We show that data obtained from a smartphone and wrist-worn device from eight seconds of movement are sufficient to detect intoxication (obtaining an AUC of 0.97).

In this paper, we make the following contributions: (1) We show that the motion sensors of smartphones and widely used wrist-worn devices can be used to identify the physiological indicators that imply intoxication (in terms of body movement) based on free gait and may provide an alternative to traditional ad hoc sensors and tests that focus on breath or blood samples. (2) We formalize the task of intoxication detection as a supervised machine learning task based on body movement measurements derived from the sensors of smartphones and wrist-worn devices.

We used an actual breathalyzer (as used by police departments) in order to label our data and train our models to evaluate our results.

The remainder of this paper is structured as follows: In Section 2, we review related works, and in Section 3, we present the proposed approach. In Section 4, we present the experiment, methodology, and ethics. In Section 5 we describe the data processing method, and we present our evaluation and results in Section 6. Finally, in Section 7 and Section 8, we conclude the paper and present future work directions, respectively.

## 2. Related Work

In this section, we review related work in the fields of gait analysis, context/activity detection using commercial wearable devices, and intoxication detection.

### 2.1. Gait Analysis

Gait analysis has been studied for many years, even before the era of wearable devices. A decade ago, researchers were already using ad hoc sensors specially designed for research purposes. Mantyjarvi et al. [4] analyzed data collected from worn accelerometer devices in order to identify subjects by their gait, Gafurov et al. [5] used a worn accelerometer for authentication and identification based on the subjects’ gait, and Lu et al. [6] showed that authentication from gait is also possible from smartphone sensors. Aminian et al. [7] analyzed accelerometer and gyroscope measurements from ad hoc sensors that they designed to be worn on a shoe in order to explore gait. Xu et al. [8] presented a novel system for gait analysis using smartphones and three sensors located within shoe insoles to provide remote analysis of the user’s gait.

### 2.2. Context and Activity Detection Using Commercial Wearable Devices

In the area of activity detection, Thomaz et al. [9] used smartwatch motion sensors in order to detect eating instances. Ranjan et al. [10] analyzed smartwatch sensors during specific home-based activities (such as turning on a light switch) to identify subjects based on hand gestures. In the field of emotion detection, Hernandez et al. [11] analyzed head movement from Google Glass motion sensors in order to detect stress, fear, and calm. Hernandez et al. [12] analyzed smartwatch motion sensors to estimate heart and breathing rates. Mazilu et al. [13] analyzed wrist movement to detect gait freezing in Parkinson’s disease using the data sensors of smart watches and wristbands, while Gabus et al. [14] and Casilari et al. [15] used a smartwatch in order to detect falls.

### 2.3. Intoxication Detection

Despite the importance of detecting intoxication, there has been a limited amount of research that addresses the domain of intoxication detection using ubiquitous technology. A recent study [16] showed that intoxication can be detected via a dedicated smartphone application that challenges the subject with various tasks, such as typing, sweeping, and other reaction tests. However this method is not passive and can be considered a software alternative to a breathalyzer, since it suffers from another shortcoming of the breathalyzer: its effectiveness is dependent on the cooperativeness of the subject. Kao et al. [17] analyzed the accelerometer data collected from subjects’ smartphones and compared the step times and gait stretch of sober and intoxicated subjects. This research was limited in scope in that it only used three subjects. In addition, it was not aimed at detecting whether a person was intoxicated based on data collected from the device; instead, the study compared differences in the gait of intoxicated and sober subjects. Arnold et al. [18] investigated whether a smartphone user’s alcohol intoxication level (how many drinks they had) can be inferred from their gait. The authors used time and frequency domain features extracted from the device’s accelerometer to classify the number of drinks a subject consumed based on the following ranges: 0–2 drinks (sober), 3–6 drinks (tipsy), or 6+ drinks (drunk). However, their methodology is not admissible, because some people do not become intoxicated from two drinks while others do, as this depends on physiological (e.g., the subject’s weight) and non-physiological factors (e.g., whether the subject has eaten while drinking). Several studies have utilized ubiquitous technology to detect intoxication based on driving patterns. Dai et al. [19] and Goswami et al. [20] used mobile phone sensors and pattern recognition techniques to classify drunk drivers based on driving patterns. Various other approaches for intoxication detection have also been investigated. Thien et al. [21] and Wilson et al. [22] attempted to simulate the HGN (horizontal gaze nystagmus) test [23] in order to detect intoxication using a camera (i.e., smartphone camera) and computer vision methods. Hossain et al. [24] used machine learning algorithms to identify tweets sent under the influence of alcohol (based on text). None of the abovementioned methods were validated against an admissible breathalyzer, and the authors did not test the accuracy of the methods on a large number of subjects.

## 3. Proposed Approach

In this section, we describe Virtual Breathalyzer, an approach for detecting intoxicated users based on free gait data obtained from a smartphone and wrist-worn device.

The short-term effects of alcohol consumption on subjects range from a decrease in anxiety and motor skills and euphoria at lower doses to intoxication (drunkenness), stupor, unconsciousness, anterograde amnesia (memory “blackouts”), and central nervous system depression at higher doses [1]. As a result, various field sobriety tests are administered by police officers as a preliminary step before a subject takes a BrAC test using a breathalyzer.

One of the most well-known field sobriety tests administered by police departments in order to detect whether a person is intoxicated is the walk and turn test in which a police officer asks a subject to take nine steps, heel-to-toe, along a straight line; turn on one foot; and return by taking nine steps in the opposite direction. During the test, the officer looks for seven indicators of impairment. If the driver exhibits two or more of the indicators during the test, there is a significant likelihood that the subject is intoxicated (according to the US National Highway Traffic Safety Administration/NHTSA [25]).

Based on the effectiveness of the walk and turn test, we suggest the following approach: detecting whether a subject is intoxicated by analyzing differences in his/her free gait. We propose identifying the physiological indicators that imply drunkenness (in terms of body movement) based on the difference between two data samples of free gait. Each sample consists of motion sensor data obtained via devices that are carried/worn by an individual.

We believe that smartphones and wrist-worn devices can be used for the purpose of intoxication detection based on free gait because (1) smartphones and wrist-worn devices are heavily adopted, with wrist-worn devices being the most commonly used and popular type of wearable device; according to a 2014 survey, one out of every six people owned a wrist-worn device [26], and a 2019 survey showed that their adoption rate increased, with 56% of people owning a wrist-worn device [27]; (2) smartphones and wrist-worn devices contain motion sensors capable of measuring free gait; and (3) most people have their smartphones and wrist-worn devices on them all the time (according to a recent survey [27]).

The first data sample consists of a standard free gait sample recorded by an individual during a time period in which they are likely to be sober (e.g., during the morning or afternoon). The second sample is recorded during the time of interest (e.g., the time the individual is believed to be intoxicated).

Using a smartphone and wrist-worn device, the free gait of an individual can be recorded both while they are sober and when they are believed to be intoxicated. By identifying features of the individual’s free gait and determining whether the differences between these features when sober and when believed to be intoxicated exceed a predetermined threshold, a trained machine learning model can determine whether the individual is intoxicated.

Algorithm 1 presents a high-level solution for detecting intoxication based on the Virtual Breathalyzer approach. It receives four inputs: a trained intoxication detection *Model*; two samples of free gait: (1) when the user is sober (*sSober*) and (2) when the user is believed to be intoxicated (*sSuspect*); and a learned *Threshold*. First, features are extracted for each sample of free gait for *fSuspect* and *fSober* (lines 7–8). Then, the difference between the features *fSuspect* and *fSober* is calculated (lines 10–12). The difference is then classified using a trained intoxication detection *Model* (line 8). Finally, the result is returned according to a learned *Threshold*.
**Algorithm 1** Is Intoxicated?1:Input:Model—Intoxication Detection Model2:Input:sSober—Gait Measurements while Sober3:Input:sSuspect—Suspected Gait Measurements4:Input:Threshold—Confidence threshold5:Output:Boolean—True/False for intoxication6:**procedure** **isIntoxicated?**7:    fSober[]=features(sSober)8:    fSuspect[]=features(sSuspect)9:    n=length(fSober)10:    difference[]=newarray[n]11:    **for** (i=0;i<n;i++) **do**12:        difference[i]=fSuspect[i]−fSober[i]13:    Probability=Model.classify(difference)14:    return(Probability>Threshold)

## 4. The Experiment and Methodology

In this section, we describe the experiments we conducted in order to evaluate whether data from a smartphone and a wrist-worn device can be used to detect if the device owner is intoxicated. We present the experimental framework we developed, the ethical considerations we had to take into account, the experimental protocol, and the methodology.

### 4.1. Experimental Framework

Most commercial wrist-worn devices are equipped with motion sensors and include an SDK to allow users to program them easily. We chose to use the Microsoft Band as a wrist-worn device in our experiment, because: (1) its SDK has clear documentation, (2) it is easy to program the device, and (3) the device has both accelerometer and gyroscope sensors, and each sample is provided over three axes (x, y, and z).

We paired the Microsoft Band to a smartphone (Samsung Galaxy S4) using Bluetooth communication. We used the Microsoft Band’s SDK to develop a dedicated application for the smartphone that sampled motion sensor data from the wrist-worn device and smartphone. The motion sensor data was sampled from the Samsung Galaxy S4 at 180 Hz and the Microsoft Band at 62 Hz and recorded as a time series in nanoseconds.

The application generated a beep sound that was played to the subject (via headphones) and triggered the subject to start walking (while wearing the devices) until the application generated a second beep 16 s later. In order to measure the subject’s gait, the application sampled the sensors for eight seconds, a time period that started on the sixth second of the experiment and continued until the fourteenth second. The stages of the experiment are presented in Figure 1.

We decided that using eight seconds of movement was the optimal way to conduct the experiment and obtain the samples for the following reasons: (1) intoxication affects a subject’s gait and balance; (2) the user may be parked a few meters away from the bar so the walk from the bar to the car may be short; (3) gait is probably the best way to ensure that the devices are carried/worn by the user instead of sitting on a desk or table (in the context of a bar); and (4) free gait measurements can be obtained from the user passively by detecting walking instances (from smartphone/wrist-worn device sensors such as the accelerometer, gyroscope, and GPS).

In addition, we purchased a Drager Alcotest 5510 breathalyzer (https://www.draeger.com/en_me/Products/Alcotest-5510, accessed on 18 April 2022) in order to obtain BrAC samples. This breathalyzer outputs results in micrograms of alcohol per liter of breath. We chose this type of breathalyzer, because it is a professional breathalyzer used by police departments around the world.

### 4.2. Ethical Considerations

The experiment involved collecting data from intoxicated and sober subjects, which was approved by the institutional review board (IRB), subject to the following precautions:(1)Only individuals that went to a bar in order to drink of their own accord could participate in the experiment; in this way, the onus for any consequences resulting from such drinking would be on the subjects.(2)Only individuals that did not drive to the bar and would not drive back from the bar could participate in the experiment.(3)Anonymization was applied to the data. At the beginning of the experiment, a random user ID number was assigned to each subject, and this user ID number served as the identifier of the subject, rather than his/her actual identifying information. The mapping between the experiment’s user ID numbers and the identity of the subjects was stored in a hard copy document that was kept in a safe box; at the end of the experiment, we destroyed this document.(4)During the experiment, the data collected were stored encrypted in the local storage of the smartphone (which was not connected to the Internet during the experiment). At the end of the experiment, the data was copied to a local server (i.e., within the institutional network), which was not connected to the Internet. Only anonymized information of the subjects was kept for further analysis.(5)Subjects were paid for their participation in the study (each subject received the equivalent of 15 USD in local currency).

### 4.3. Methodology

In order to sample as many people as possible, our experiment took place on three evenings at three bars that offer an ”all you can drink” option (we visited one bar each evening). The Google Maps locations of the bars are provided (https://www.google.com/maps/place/Shlomo+Ibn+Gabirol+St+13,+Tel+Aviv-Yafo/, accessed on 18 April 2022), (https://www.google.com/maps/place/Shlomo+Ibn+Gabirol+St+17,+Tel+Aviv-Yafo/, accessed on 18 April 2022), (https://www.google.com/maps/place/Shlomo+Ibn+Gabirol+St+33,+Tel+Aviv-Yafo/, accessed on 18 April 2022). We waited for people to arrive at the bars, and just before they ordered their first drink, we asked them to participate in our research (participation entailed providing a gait sample during two brief experimental sessions with a smartphone and wrist-worn device, as well as providing two breath samples a few seconds before the sessions started). We explained that they would receive the equivalent of 15 USD in local currency for their participation. We also told the subjects that they would be compensated even if they chose not to drink at all, so drinking was not obligatory. Each subject signed a document stating that he/she came to the bar in order to drink of his/her own accord and that he/she did not drive to the bar and would not drive from the bar (as we were instructed by the IRB). The breathalyzer was calibrated at the beginning of each evening according to the manufacturer’s instructions.

The experiment was conducted in two sessions. The first session took place before the subjects had their first drink. The second session took place at least 15 min after the subject’s last drink, just before they intended to leave the bar. We consulted with police authorities regarding the breathalyzer test, and they told us to wait 15 min after the subject had their last drink in order to obtain an accurate BrAC specimen. During each session, our subjects provided us with a gait sample and a BrAC specimen. Their gait was recorded using the application that we developed (described at the beginning of this section). The BrAC specimen was measured with the breathalyzer; the result was used to label each gait sample.

Our subjects were outfitted with the devices as follows: they were asked to wear the Microsoft Band on their left or right wrist (at their discretion) and carry a smartphone in a rear pocket (as can be seen in Figure 2).

Each subject also wore headphones that were used to hear the beeps used to indicate when they should start/stop walking. Thirty subjects participated in our study, each of whom was instructed to walk (while wearing the devices) in any direction they wished until they heard a beep in the headphones, as can be seen in Figure 1. Each subject provided two free gait samples, one before and one after drinking, resulting in the 60 free gait samples collected in the field experiment.

## 5. Processing the Data

In the following section, we describe the extracted features and the process of creating the dataset.

### 5.1. Feature Engineering

Differences in walking caused by intoxication are expressed as difficulty walking in a straight line and maintaining balance, and swaying. These indicators appear even with the consumption of a small amount of alcohol and can be detected by police officers in the field sobriety test (walk and turn test) without a dedicated device. The walk and turn test is usually performed by officers before a breathalyzer test in order to save the long process of obtaining a breath sample from individuals that are not shown to be intoxicated based on the field sobriety test.

Since we used data obtained from motion sensors, we extracted features that can be informative as a means of detecting the abovementioned gait differences. The first type of features that we used are features from the spectrum domain. Previous studies demonstrated the effectiveness of extracting such features from motion sensors [4,12]. We applied a fast Fourier transform (FFT), and extracted features that represent the distribution of the power of the signals across the spectrum domain by taking the average power for four ranges in the spectrum. Such features may indicate physiological changes resulting from alcohol consumption that are associated with reduced frequency of movement as a result of difficulty in maintaining balance while walking. We extracted four features for each axis (x, y, z), each device (smartphone and wrist-worn device), and each device sensor (gyroscope and accelerometer). In total, we extracted 48 such features.

The second type of features that we used are statistical features. Previous studies demonstrated the effectiveness of extracting such features from motion sensors [6,9]. We extracted five features that represent high-level information about the signals. Such features may indicate physiological changes associated with intoxication, such as decreased average acceleration as a result of difficulty maintaining balance. We extracted features for each axis, each device, and each device sensor. In total, we extracted 60 such features.

The third type of features used were histogram features. We presented the signals as histograms, as done in previous studies [28,29]. We extracted a histogram that represents the distribution of the values of the signals across the time domain between the minimum and maximum values. Such features may indicate differences in the patterns of movement (and specifically, the distribution of the movement) as a result of the abovementioned indicators. We extracted six features for each axis, each device, and each device sensor. In total, we extracted 72 such features.

Finally, we extracted known gait features that have been shown to yield good results in previous studies [30,31]. We extracted four features (zero crossing rate, mean crossing rate, median, and RMS). These features may indicate differences in the characteristics of a person’s gait that are the result of difficulty walking. We extracted features for each axis, each device, and each device sensor. In total, we extracted 48 such features.

In total, 228 features were extracted and utilized for our method.

### 5.2. Creating the Dataset

As mentioned in Section 4, each subject contributed two breath specimens and gait samples (obtained in two sessions—before and after drinking). Each gait sample is comprised of sensor readings (measurements) obtained from a smartphone and wrist-worn device. The accelerometer and gyroscope were sampled from the smartphone and wrist-worn device.

Given person *p* and his/her two gait samples: *s-before* (measurement taken before alcohol consumption) and *s-after* (measurement taken after alcohol consumption), we processed the samples as follows:(1)Feature Extraction—We extracted two feature vectors: the *f-before* vector (extracted from *s-before*) and the *f-after* vector (extracted from *s-after*).(2)Difference Calculation—We calculated a new feature vector called the *f-difference*. These features represent the difference (for each feature) between the *f-after* and *f-before* values. The difference signifies the effects of alcohol consumption on the subject’s movement and is calculated by subtracting each of the features in *f-before* from its correlative feature in *f-after*.(3)Labeling—We labeled the sample of each subject as intoxicated/sober according to the result of the professional breathalyzer for known BrAC thresholds.

The dataset creation process resulted in 30 labeled instances extracted from 30 users, representing the differences between the extracted features before and after drinking. We used these data to train supervised machine learning models for intoxication detection. We analyzed the data as a classification task, with the goal of determining whether a person is intoxicated or sober according to known BrAC thresholds as measured using a breathalyzer. More precisely, we aimed to train a model that determines whether a person is intoxicated or not using differences in the subject’s gait features. We chose to classify our instances according to three common BrAC thresholds: 220, 240, and 380. These BrAC thresholds are commonly used by countries around the world (see Table 1). We consider an instance labeled by a breathalyzer result (BrAC) to be sober if its value is less than the threshold and intoxicated if its value exceeds the threshold.

The breakdown of the subject’s sober/drunk states according to the common BrAC thresholds 220, 240, and 380 is presented in Figure 3. At the lower BrAC thresholds of alcohol concentration (220, 240), the data is distributed such that 20–33% of the total number of subjects were considered intoxicated. At the highest threshold (380), 10% of the subjects were considered intoxicated.

## 6. Evaluation

In this section, we describe the algorithms used and the evaluation protocol. In addition, we report the performance of the intoxication detection method specified by Algorithm 1 using the models that we trained.

### 6.1. Algorithms & Evaluation Protocol

Five different machine learning models were evaluated to allow for a versatile yet comprehensive representation of the model’s performance. The first model that we evaluated was Naive Bayes which belongs to a family of simple probabilistic classifiers. The second model evaluated was Logistic Regression. This model is able to obtain good results in cases where the two classes can be adequately separated using a linear function. The third model used was Support Vector Machines which is used to identify the maximum margin hyperplane that can separate classes. Finally, we evaluated two ensemble-based classifiers: Gradient Boosting Machine (GBM) and AdaBoost. GBM trains a sequence of trees where each successive tree aims to predict the pseudo-residuals of the preceding trees. This method allowed us to combine a large number of classification trees with a low learning rate. AdaBoost trains a set of weak learners (decision trees) and combines them into a weighted sum that represents the final outcome.

Since our data is based on samples from 30 subjects, we could utilize the leave-one-user-out protocol, i.e., the learning process was repeated 30 times, and in each test, 29 subjects were used as a training set, and one subject was used as a test set to evaluate the predictive performance of the method. The leave-one-user-out protocol allowed us to evaluate the performance of the suggested method by utilizing the entire set of instances in the data for training and evaluation. We report the following metrics: area under the receiver operating characteristic curve (AUC), false positive rate (FPR), and true positive rate (TPR). The results that we report in this section are the average of 30 models that were trained and evaluated on the dataset for each task.

### 6.2. Results

We use Algorithm 1 in order to evaluate the following:(1)our method’s performance according to various BrAC thresholds;(2)our method’s performance when using various detection policies; and(3)the importance of each device, sensor, and set of features in terms of the method’s performance.

#### 6.2.1. Performance with Various BrAC Thresholds (220, 240, 380 BrAC)


**ROC/AUC Results**


We start by assessing the performance of the intoxication detection method from data obtained from a smartphone and a wrist-worn device. Table 2 presents the AUC results for each of the classification models for BrAC thresholds of 220, 240, and 380. As can be seen, the GBM and AdaBoost classifiers yielded high accuracy rates for these thresholds. Figure 4 and Figure 5 present the ROC curves for the AdaBoost and Gradient Boosting classifiers.


**Classification Accuracy**


We also analyzed the classifiers’ decisions. The confusion matrices for the AdaBoost and Gradient Boosting classifiers for BrAC thresholds of 220, 240, and 380 are presented in Table 3 and Table 4. As can be seen, for the threshold of 380 BrAC every subject is classified as sober, demonstrating a difficulty with detecting intoxication for this BrAC threshold. This can be explained by the highly imbalanced dataset, with most subjects (90%) labeled as sober due to the high BrAC threshold.

#### 6.2.2. Performance with Various Detection Policies

Here, we set out to evaluate the performance of the intoxication detection method according to two policies. Table 3 and Table 4 present misclassifications (FNR and FPR) for BrAC thresholds of 220, 240, and 380.


**Each Intoxicated Subject is Classified as Intoxicated (0 FNR)**


Misclassifying a drunk user as sober would provide false confidence to a user, implying that they are not intoxicated, which could cause them to perform risky behavior, such as driving while unknowingly intoxicated. In order to avoid such incidents, we wanted to test the performance of a model on a policy whereby each intoxicated subject is predicted as intoxicated. In order to do so, we fixed the TPR at 1.0 (the true class is intoxicated) and assessed the impact of this limitation on the FPR, i.e., we looked at the percentage of sober subjects that were misclassified as intoxicated.

Table 5 presents the FPR results of the Gradient Boosting and AdaBoost classifiers for BrAC thresholds of 220, 240, and 380. As can be seen from the results, applying a constraint of detecting all intoxicated subjects caused up to 30% of the sober subjects to be misclassified as intoxicated for BrAC thresholds of 220, 240, and 380.


**Each Subject Classified as Intoxicated is Actually Intoxicated (0 FPR)**


We also evaluated the performance of the method on another policy whereby each intoxicated subject that is classified as intoxicated by the method is actually intoxicated in reality. In order to do so, we fixed the FPR at zero (the positive class is intoxicated) and assessed the impact of this limitation on the TPR, i.e., we looked at the percentage of intoxicated subjects that were misclassified as sober.

Table 6 presents the TPR results of the Gradient Boosting and AdaBoost classifiers for BrAC thresholds of 220, 240, and 380. As can be seen from the results, the impact of applying a constraint of detecting only intoxicated subjects is that this approach is only effective for a BrAC threshold of 220, since 40–55% of the intoxicated subjects are detected (when using a Gradient Boosting classifier as the intoxication detection model). However, for all other BrAC thresholds, all of the intoxicated subjects are misclassified.

#### 6.2.3. Importance of Devices, Features, and Sensors Regarding Performance

In this experiment, we aimed to determine the impact of each device, sensor, and set of features on the performance. 


**Importance of Devices**


We started by evaluating the performance for data that was obtained exclusively from a smartphone or a wrist-worn device. We trained AdaBoost and Gradient Boosting classifiers with data obtained from a single device for BrAC thresholds of 220, 240, and 380.

Table 7 presents the results of the AdaBoost and Gradient Boosting classifiers for data obtained from a smartphone, wrist-worn device, and both devices (for comparison). As can be seen from the results, measurements of movements from both devices are required to accurately classify a subject as intoxicated/sober.


**Importance of Features**


In the feature extraction process, we extracted four types of features. Since the gait of individuals changes as a result of alcohol consumption, we wanted to identify the best set of indicators to detect intoxication (based on body movement patterns) and determine which of the following is most effective at this task: the distribution of the movement (histogram), frequency of the movement, statistical features, or known gait features.

In order to do so, we used the dataset and trained Gradient Boosting and AdaBoost classifiers for BrAC thresholds of 220, 240, and 380. We classified each instance using two methods. The first classification method used a specific set of features among the sets (histogram, known gait features, frequency features (FFT), statistical features). The second classification method used all of the other sets of features (except the set used in the first method). Figure 6 presents the average AUC results for BrAC thresholds of 220, 240, and 380. As can be seen from the results, the models that were trained on only statistical features outperformed the models that were trained without them. All other models that were trained on a certain set of features were unable to obtain higher scores than the models that were trained without them. However, models trained with a combination of features (such as every set except FFT) achieved higher performance. From this, we conclude that a combination of the entire set of features is required to train an effective/accurate intoxication detection model.


**Importance of Sensors**


Finally, we examine the impact of data from each sensor on the results. In order to do so, we followed the same protocol used to test the feature robustness: we trained Gradient Boosting and AdaBoost classifiers for BrAC thresholds of 220, 240, and 380. We classified each instance using a model that was only trained on accelerometer features and a model that was only trained on gyroscope features. Figure 7 presents the average AUC results for BrAC thresholds of 220, 240, and 380. As can be seen from the results, a model that was trained only on accelerometer measurements yields nearly the same results as a model trained on measurements from both sensors. Given this, we conclude that subjects’ acceleration when walking is highly informative in the detection of intoxication.

## 7. Conclusions and Discussion

In this paper, we present Virtual Breathalyzer, a novel approach to detect intoxication using data from the motion sensors of commercial wearable devices which may be used as an alternative by users when a breathalyzer is not available. We conducted an experiment involving 30 patrons from three different bars to evaluate our approach. Our experiment demonstrated the proposed approach’s ability to accurately detect intoxication using just a smartphone and wrist-worn device. An AUC of 0.97 was obtained for a BrAC threshold of 240 micrograms of alcohol per one liter of breath. Using two simple gait samples (from a car to a bar and vice versa), a system based on this approach can be used to prevent people from driving under the influence of alcohol and could also be used to trigger the device owner’s connected car to prevent ignition in cases in which the owner is detected as drunk.

The significance of the Virtual Breathalyzer approach with respect to the methods proposed in related work is that our approach: (1) requires minimal/no cooperation on the part of the subject (unlike [16]), (2) utilizes ubiquitous, commercial device sensors for detecting intoxication rather than ad hoc sensors (unlike a blood or breath test) (3) is validated against the results of an admissible police breathalyzer (unlike previous methods [19,20,21,22,24]), and (4) can be utilized in real time to prevent a user from driving while intoxicated.

Some might argue that intoxication detection via wearable devices provides a welcome opportunity to notify a device owner that they are intoxicated in order to prevent them from driving under the influence of alcohol. Others might argue that intoxication detection via wearable devices threatens people’s privacy, because it could be exploited as a means of learning about the habits of the device owner (e.g., which could lead an employer to fire an employee due to his/her drinking habits) or to prove that a device owner has driven under the influence of alcohol. The main objective of this research was to show that motion sensors can be used as alternative to the traditional blood and breath tests for intoxication detection, rather than taking a particular side in an argument about the advantages and disadvantages of such a method.

## 8. Future Work

There are numerous opportunities to extend this work:(1)Deriving additional insights via alternative virtual, passive methods: additional research is needed to detect intoxication/drug use indirectly via passive and virtual methods. For example, the physiological indicators (e.g., sweat, reduced movement) associated with drug use might also be identified via wearable device sensors (skin conductivity and motion sensors).(2)Deriving insights from aggregated/low resolution data: additional research is also required in order to derive insights from aggregated data. For example, a recent study [30] compared the effectiveness of various statistical features used to detect a subject’s gait from wearable devices. The ability to derive insights from aggregated data can enable virtual intoxication detection methods to be used to make inferences about an individual’s cognitive state.(3)Data quality: additional research is required to understand whether the quality of the data obtained by the sensors of commercial wearable devices can replace dedicated sensors for general health/status inference. For example, cardiovascular data obtained from a dedicated sensor can be used to detect lies [32]; however a recent study revealed that the cardiovascular data obtained from an Apple watch generates false alarms 90% of the time for pulse readings that are associated with a patient’s cardiac condition [33]. We believe that additional research is also required to explore the accuracy and errors of the sensors that are integrated in wearable devices.(4)Dimensionality Reduction and Feature Analysis: additional research on performing dimensionality reduction on extracted features, as well as identifying individual features which contribute most to performance, would be a significant addition to this research in analyzing the impact of extracted features on the performance of our method.(5)General population diversity: additional research is needed in analyzing the diverse medical, geographic, and demographic variations between populations and its effect on the performance of our method, before it can be successfully deployed for general use.

## Figures and Tables

**Figure 1 sensors-22-03580-f001:**
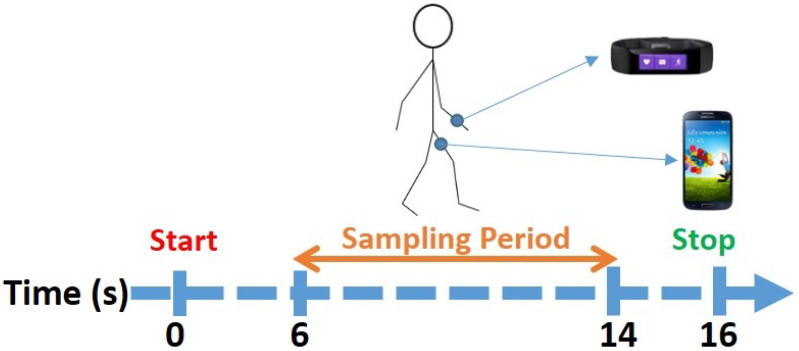
Experimental protocol: a sample of eight seconds of motion sensor data obtained when the subject was walking.

**Figure 2 sensors-22-03580-f002:**
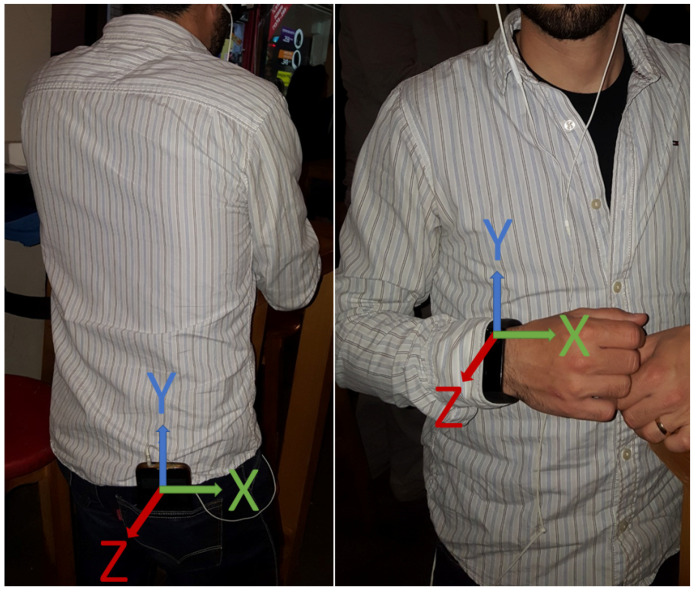
A subject outfitted with a Microsoft Band and Samsung Galaxy S4.

**Figure 3 sensors-22-03580-f003:**
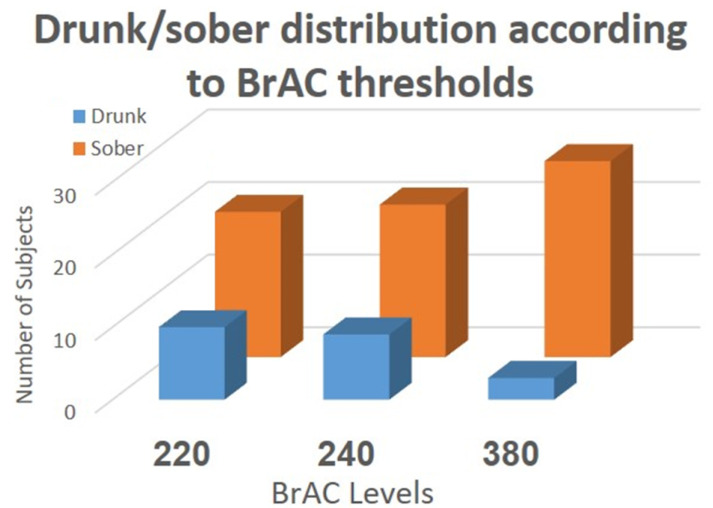
A breakdown of the subjects’ state (sober/drunk) at various BrAC levels.

**Figure 4 sensors-22-03580-f004:**
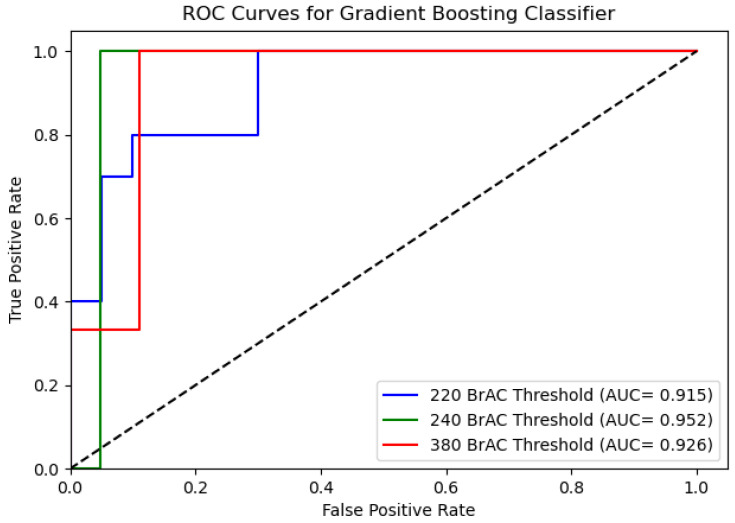
ROC curve of the Gradient Boosting classifier for BrAC thresholds of 220, 240, and 380 from data that was obtained from a smartphone and wrist-worn device.

**Figure 5 sensors-22-03580-f005:**
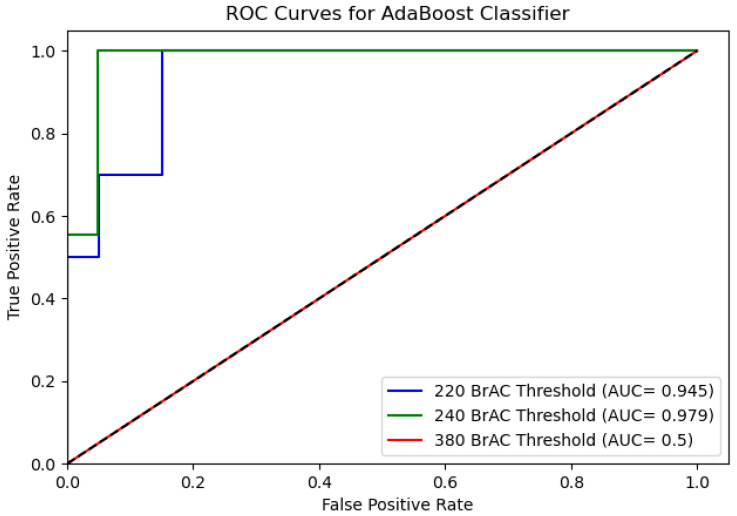
ROC curve of the AdaBoost classifier for BrAC thresholds of 220, 240, and 380 from data that was obtained from a smartphone and wrist-worn device.

**Figure 6 sensors-22-03580-f006:**
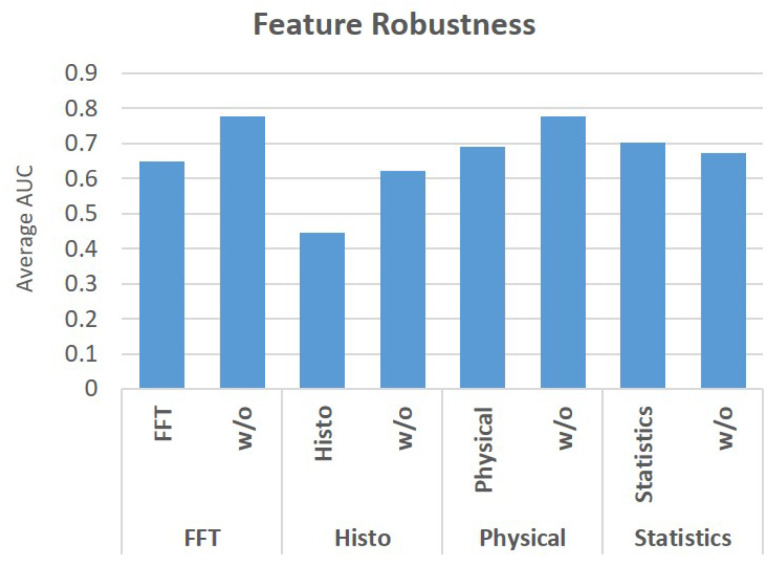
Average AUC results of the AdaBoost and Gradient Boosting classifiers based on specific types of features and without them.

**Figure 7 sensors-22-03580-f007:**
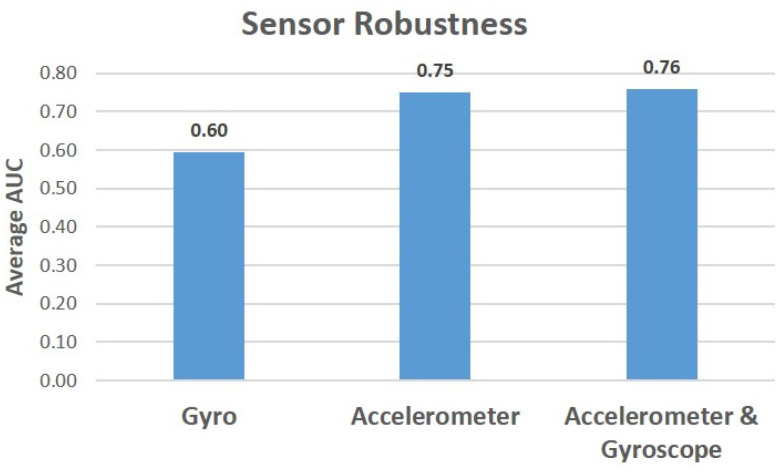
Average AUC results of the AdaBoost and Gradient Boosting classifiers based on measurements that were obtained from a single sensor and from both sensors.

**Table 1 sensors-22-03580-t001:** BrAC thresholds for intoxication around the world.

BrAC Threshold	Countries
220	Scotland, Finland, Hong Kong
240	Slovenia, South Africa, Israel
380	Malawi, Namibia, Swaziland

**Table 2 sensors-22-03580-t002:** AUC of classification algorithms: AdaBoost, Naive Bayes (NB), Linear Regression (LR), Support Vector Machines (SVM), and Gradient Boosting (GB) for BrAC thresholds of 220, 240, and 380.

	Thresholds		
	220	240	380
AdaBoost	0.945	0.979	0.500
GB	0.915	0.952	0.926
LR	0.560	0.577	0.457
NB	0.290	0.196	0.414
SVM	0.500	0.500	0.500

**Table 3 sensors-22-03580-t003:** Confusion matrices of the Gradient Boosting classifier for BrAC thresholds of 220, 240, and 380.

	Predicted					
	220		240		380	
	Drunk	Sober	Drunk	Sober	Drunk	Sober
Drunk	6	4	9	0	0	3
Sober	1	19	2	19	0	27

**Table 4 sensors-22-03580-t004:** Confusion matrices of the AdaBoost classifier for BrAC thresholds of 220, 240, and 380.

	Predicted					
	220		240		380	
	Drunk	Sober	Drunk	Sober	Drunk	Sober
Drunk	8	2	9	0	0	3
Sober	3	17	1	20	0	27

**Table 5 sensors-22-03580-t005:** Detecting all intoxicated subjects: FPR (false positive rate) of classifiers with a fixed TPR (true positive rate) of 1.0.

Thresholds	220	240	380
GBC	0.3	0.09	0.11
AdaBoost	0.15	0.04	0

**Table 6 sensors-22-03580-t006:** Detecting an intoxicated instance with no errors: TPR (true positive rate) of classifiers with a fixed FPR (false positive rate) of zero.

Thresholds	220	240	380
GBC	0.4	0	0
AdaBoost	0.4	0.55	0

**Table 7 sensors-22-03580-t007:** AUC results of the AdaBoost and Gradient Boosting classifiers based on data obtained from a smartphone, wrist-worn device, and both devices.

	Thresholds		
	220	240	380
	Gradient Boosting		
Smartphone	0.74	0.38	0.46
Wrist-Worn Device	0.52	0.68	0.92
Both	0.915	0.952	0.926
	AdaBoost		
Smartphone	0.75	0.57	0.59
Wrist-Worn Device	0.33	0.73	0.5
Both	0.945	0.979	0.5

## Data Availability

The data are not publicly available due to privacy concerns regarding the subjects involved in the study.

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
