# Peer review of "Virtual Breathalyzer: Towards the Detection of Intoxication Using Motion Sensors of Commercial Wearable Devices"

_sensors, 2022, doi:10.3390/s22093580_

Round 1

Reviewer 1 Report

Dear Authors,

The topic of the manuscript is highly interesting and has the potential of benefiting the readers of Sensors. The research conducted is scientifically sound and the results are mostly clearly presented.

There are a couple of minor issues that should be resolved. I state these in order of appearance.

However stated later on in the manuscript, it is not evident from the abstract from how many subjects were the gait samples obtained. You might consider including this information.

In general, the structure of the manuscript should be improved: significance of the method with respect to related studies is probably more appropriate for the Discussion/Conclusions section than for Proposed Approach; ‘Feature engineering’, ‘Creating the dataset’, and ‘Algorithms and evaluation’ are all part of the methodology and should be separated from results; lines 262-265 are introductory and should be stated only in the Introduction. Table 1 and Figure 3 should appear closer to the manuscript text they relate to. The Results section should be organized with more attendance.

The coordinate system axes x, y, and z in Figure 2 should be denoted clearer (should not overlap with the axis arrow’s and in general should be presented with more attendance).

If ‘Feature engineering’, the exact number of features per category should be stated. In addition, spectrum features are mentioned but they should also be more precisely elaborated. It is not clear how exactly the power spectrum distribution is considered. It is also not clear whether the features were extracted from motion signal magnitudes or from projections on coordinate system axes.

Have you considered dimensionality reduction and identifying specific features (as oppose to feature categories) that contribute the most to performance?

Reviewer 2 Report

This paper presented a novel approach for detecting intoxication from the measurements obtained by the sensors of smartphones and wrist-worn devices. The description of the content is very clear, and the experimental design is rational. Also, the author identified three directions to extend this work. Thus, I recommend its publication in its current form.

Reviewer 3 Report

Here the authors present a Virtual Breathalyzer, a novel approach to detect intoxication using data from the motion sensors of commercial wearable devices, which may be used as an alternative by users when a breathalyzer is not available. The new methodology is fancy but as the authors claim can be used as an alternative. this methodology can not be properly applied in general population due other medical and geographic condition. Perhaps, in my view this paper was well done and brights a new ideia of how you can technology can provide a wearable deviceswhich provides a welcome opportunity to notify a device owner that they are intoxicated in order to prevent them from driving under the influence of alcohol.
